# Volumetric Three-Dimensional Evaluation of the Pharyngeal Airway After Orthognathic Surgery in Patients with Skeletal Class III Malocclusion

**DOI:** 10.3390/diagnostics15172217

**Published:** 2025-09-01

**Authors:** Aslihan Zeynep Oz, Hakan El, Abdullah Alper Oz, Juan Martin Palomo

**Affiliations:** 1Department of Orthodontics, Faculty of Dentistry, Ondokuz Mayıs University, Samsun 55139, Turkey; 2Private Practice, Ankara 06510, Turkey; hakanel@gmail.com; 3Department of Orthodontics, Faculty of Dentistry, Galata University, Istanbul 34421, Turkey; alperoz@hotmail.com; 4Department of Orthodontics, School of Dental Medicine, Case Western Reserve University, Cleveland, OH 44106, USA; palomo@case.edu

**Keywords:** orthognathic surgery, Class III malocclusion, pharyngeal airway, three-dimensional imaging

## Abstract

**Background:** Orthognathic surgery significantly alters the dimensions of the pharyngeal airway. This study’s objective was to assess alterations in the pharyngeal airway volume via cone-beam computed tomography (CBCT) after orthognathic surgery in patients with skeletal Class III malocclusion. **Methods:** This retrospective study analyzed CBCT images from 23 patients with skeletal Class III malocclusion (13 females, 10 males), who were categorized into two groups based on the surgical approach: double-jaw and single-jaw surgery. The double-jaw group included 13 patients who underwent bilateral sagittal split osteotomy (BSSO) and Le Fort I osteotomy, whereas the single-jaw group included of 10 patients who had underwent BSSO only. CBCT images were evaluated both before surgery and at a minimum of three months after surgery. The oropharyngeal volume (OP), nasopharyngeal volume (NP), total airway volume, posterior airway space (PAS), and the most constricted area at the base of the tongue (minAx) were measured. Statistical analyses were performed using either paired *t*-tests or Wilcoxon signed-rank tests depending on data normality, with a significance level set at *p* < 0.01. **Results:** In the double-jaw group, a significant volumetric increase was observed in the nasopharynx (5316 ± 1948 mm^3^ to 6064 ± 1899 mm^3^; *p* = 0.010) and oropharyngeal volume decreased from 17,097 ± 5675 mm^3^ to 14,290 ± 5835 mm^3^; however, this reduction was not statistically significant (*p* = 0.017). In contrast, the single-jaw group showed a significant reduction in oropharyngeal volume from 15,620 ± 5040 mm^3^ to 12,444 ± 4701 mm^3^ (*p* = 0.010), with no significant change in nasopharyngeal volume (*p* = 0.551). Total airway volume significantly decreased only in the single-jaw group (from 20,452 ± 7754 mm^3^ to 16,846 ± 6529 mm^3^, *p* = 0.010). Additionally, both groups exhibited marked decreases in PAS and minimum axial area values (all *p* < 0.01). **Conclusions:** Orthognathic surgery led to a significant volumetric increase in the nasopharynx in the double-jaw group, whereas the oropharynx volume significantly decreased only in the single-jaw group. Additionally, both surgical approaches resulted in a marked reduction in PAS and minimum axial area values, highlighting a notable impact on posterior airway dimensions.

## 1. Introduction

The Class III skeletal pattern is characterized by anteroposterior incompatibility due to an insufficient upper jaw, an excessive lower jaw, or both. The treatment of Class III malocclusion can be performed with growth modification or camouflage treatment, depending on the patient’s growth period and the severity of the malocclusion. Orthognathic surgery in combination with traditional orthodontic treatment is required for the treatment of skeletal Class III malocclusion in adults, to improve facial aesthetics and occlusion [1,2]. Orthognathic surgery involves procedures such as osteotomies and jaw repositioning to correct bone discrepancies, as well as significant soft tissue changes [3]. Repositioning the muscles attached to the jaws and pharyngeal walls alters the volume of the pharyngeal airway; the forward or backward movement of the facial skeleton causes the airway walls to expand or contract [4,5]. Therefore, the type, direction, and magnitude of skeletal movements affect the dimensions of the pharyngeal airway [6].

A significant drawback of mandibular setback is its impact on oropharyngeal morphology, leading to airway constriction and a subsequent reduction in the posterior airway space (PAS) [6,7]. This occurs because the posterior repositioning of the mandible displaces the tongue base and associated soft tissues posteriorly, which may narrow both the oropharyngeal and hypopharyngeal airway segments [4]. The pharyngeal airway is anatomically divided into three distinct regions: the nasopharynx, oropharynx, and hypopharynx. The nasopharynx is the most superior region, located posterior to the nasal cavity and extending from the base of the skull to the level of the hard palate or posterior nasal spine (PNS). The oropharynx is inferior to the nasopharynx and extends from the palatal plane (ANS–PNS) to the level of the epiglottis or the anterior-inferior point of the second cervical vertebra (C2). The hypopharynx begins below the oropharynx and continues to the epiglottis plane. Each of these regions has unique anatomical and functional characteristics, and changes in skeletal morphology due to orthognathic surgery may differentially affect their volumetric dimensions [8]. Each of these regions can be affected differently by the surgical repositioning of the jaw. While mandibular setback may decrease the oropharyngeal and hypopharyngeal volumes by pushing the tongue and hyoid bone backward, maxillary advancement has the opposite effect: it increases the nasopharyngeal and upper oropharyngeal airway spaces by moving the maxilla and soft palate anteriorly, thereby enlarging the retropalatal area [5,9]. In addition, genioplasty, particularly when involving the anterior repositioning of the chin, can positively influence airway patency by altering the tension and position of the suprahyoid muscles [10]. These opposing effects are particularly relevant in bimaxillary orthognathic surgery, where mandibular setback is combined with maxillary advancement. Studies have shown that when sufficient maxillary advancement is performed, it may mitigate or even reverse the narrowing effects of mandibular setback, resulting in a neutral or slightly favorable impact on the total airway volume [7,11].

Interest in the study of the PAS after orthognathic surgery has increased in recent years [4,8,12]. The primary reason for this interest is the tendency to decrease in airways after surgery, resulting in healthy patients experiencing symptoms related to sleep apnea, particularly in cases of a Class III deformity [13]. A decrease in airway volume and narrowing that prevents air passage are important risk factors for the emergence of obstructive sleep apnea (OSA) [14]. These findings underscore the importance of individualized surgical planning and volumetric airway assessment, particularly in patients at risk of obstructive sleep apnea or those with compromised respiratory function. Patient-related characteristics such as age, gender, body mass index (BMI), preoperative airway dimensions, and positioning, could affect soft tissue behavior and airway dimensions [15,16]. Importantly, these variables should be considered when evaluating airway outcomes.

Most studies have evaluated airway dimensions on cephalometric films [12,17]. However, assessing a three-dimensional (3D) structure in a two-dimensional (2D) form has limitations [18]. In recent years, the number of studies evaluating the airway three-dimensionally has increased with the development of cone-beam computed tomography (CBCT) devices, alternatives with low radiation doses and costs. In comparison to medical CT, CBCT enables airway imaging with the patient in a seated posture, resulting in quicker acquisition times and decreasing the chance of patient movement, which can impact volumetric measurements [19]. Lenza et al. [20] demonstrated how CBCT-based 3D analysis can clarify upper airway characteristics in comparison to single linear measurements conducted on cephalograms.

Many studies have reported a decrease in airway size following mandibular setback surgery, especially in the oropharyngeal and hypopharyngeal regions [21,22,23]. When comparing single-jaw versus bimaxillary surgery, several authors report that bimaxillary surgery causes less airway reduction, likely due to the counterbalancing effect of maxillary advancement [12,24,25,26]. However, other studies argue that orthognathic surgery does not consistently alter airway dimensions, potentially due to heterogeneity in study designs, surgical techniques, or imaging time points [27]. For instance, some studies included patients with maxillary impaction or mandibular asymmetry, which may influence airway behavior differently [28,29]. Moreover, a previous meta-analysis concluded that evidence regarding volumetric airway changes after orthognathic surgery remains inconclusive, partly due to the inconsistent measurement protocols used [30]. Despite the growing body of literature on CBCT and 3D segmentation tools, no consensus exists on the extent and regional distribution of airway changes, highlighting the need for more standardized and segmented volumetric evaluations, such as the current study [31,32].

Based on previous studies on postoperative airway enlargement, we hypothesized that orthognathic surgery would result in a significant increase in pharyngeal airway volume, particularly the oropharyngeal region, in individuals with Class III malocclusion. The objective of our study was to assess segmental changes in pharyngeal airway volume following orthognathic surgery in individuals with Class III malocclusion and severe skeletal discrepancies using three-dimensional CBCT. To our knowledge, this study represents one of the most recent volumetric analyses focusing specifically on nasopharyngeal and oropharyngeal regions in this patient population.

## 2. Materials and Methods

In our retrospective study, records of 23 skeletal class III patients, 13 females and 10 males were used. The mean preoperative age was 21.3 years with an overall age range of 18–34 years. According to the type of orthognathic surgery, patients were divided into two groups. In total, 13 patients (8 female and 5 male) underwent double-jaw surgery (bilateral sagittal split osteotomy (BSSO) and LeFort 1 surgery), and 10 patients (4 female, 6 male) were treated with single-jaw surgery (BSSO only). Patients were included who had CBCT images captured before surgery (T1) and at a minimum of three months post-surgery (T2). The inclusion criteria for Cl III patients were as follows: no presence of craniofacial abnormalities, no severe facial asymmetry, and no respiratory problems. The exclusion criteria were as follows: history of upper airway surgery, craniofacial syndromes, and diagnosed sleep apnea. The power analysis was conducted based on the preoperative mean airway volume values reported by Agarwal et al. [33], using G*Power version 3.1.9.4. The minimum number of subjects required was determined as 20, 10 for each group. (Type I Error = 0.05; Power of Test 0.85; Effect size 1.28). Approval for this retrospective study was provided by the Ondokuz Mayıs University Clinical Research Ethics Committee (Decision date: 12 August 2016, No: 2016/283). This retrospective cross-sectional study was conducted in accordance with the principles of the Declaration of Helsinki. Written informed consent was obtained from all patients, allowing their data and radiological findings to be used for future research.

CBCT recordings were conducted using the Galileos Comfort Plus (Sirona Dental Systems Inc., Bensheim, Germany) device at 98 kV and 25 mA; a scanning area with a diameter of 15.4 cm was captured with a scan time of 14 s and an isotopic voxel size of 0.25. Patients were positioned in the upright position with the Frankfort horizontal plane parallel to the floor to ensure a natural head position.

All CBCT scans were transferred to the InVivo 5.3 (Anatomage, San Jose, CA, USA) software program using the digital imaging and communications in medicine images (DICOMs) format. The line extending from the Sella to the posterior nasal spine defines the upper margin of the nasopharynx (NP); the palatal plane (ANS-PNS) is delineated by the lower margin of the NP and upper margin of the oropharynx (OP). The lower margin of the oropharynx (OP) was formed by a line drawn parallel to the palatal plane from the lowest and anterior points of the second cervical vertebra (Figure 1). In addition to volume measurements, the posterior airway space (PAS), which is the narrowest space at the base of the tongue and is limited by soft tissue, and the narrowest area at the base of the tongue (minAx), were measured (Figure 2). Segmentation was semi-automated, with manual adjustments made when necessary to ensure accuracy. The airway was segmented based on anatomical landmarks, with the nasopharynx, and oropharynx boundaries defined according to established cephalometric references. All measurements were made by an experienced operator (HE), who was blinded to the surgical groups. To assess intra-observer reliability, measurements from 10 randomly selected patients were repeated after a two-week interval. The intraclass correlation coefficient (ICC) values for airway volume, PAS, and minAx ranged from 0.91 to 0.96, demonstrating excellent reliability.

### Statistical Analysis

The data obtained were analyzed with the SPSS 22.0 statistical program. The normality of the data was assessed using the Shapiro–Wilk test. For within-group comparisons at baseline and the end of treatment, the paired *t*-test was used for normally distributed data, and the Wilcoxon signed-rank test was used for non-normally distributed data. For between group comparisons, the differences between airway measurements at the end and beginning of treatment were used. Normally distributed data were analyzed via the independent *t*-test, and non-normally distributed data were analyzed using the Mann–Whitney U test. Given the multiple comparisons across three airway regions (nasopharynx, oropharynx, and total airway), the Bonferroni correction was applied to control for type I error, resulting in an adjusted significance level of *p* < 0.01. In addition to *p*-values, effect sizes were calculated using Cohen’s d for normally distributed variables to assess the magnitude of these differences. According to Cohen’s guidelines, *d* values of 0.2, 0.5, and 0.8 indicated small, medium, and large effects, respectively. These values provide insight into the clinical relevance of the volumetric changes. The associations between the minAx, PAS, and total airway volume were evaluated using Spearman’s rank correlation coefficient, due to the non-normal distribution of the data.

## 3. Results

In the double-jaw group, the average forward movement of the maxilla was 5.25 mm, and the setback amount of the mandible was 7.75 mm. By comparison, in the single-jaw group, the average setback of the mandible was 7.43 mm (Table 1).

Measurements of one patient from each group were shown in Figure 3 for better understanding of the results.

Table 2 shows the volumes of the oropharynx, nasopharynx, and total airway before and after orthognathic surgery. In the double-jaw group, oropharyngeal volume decreased from 17,097 ± 5675 mm^3^ at T1 to 14,290 ± 5835 mm^3^ at T2; however, this reduction was not statistically significant (*p* = 0.017). In contrast, nasopharynx volume significantly increased from 5316 ± 1948 mm^3^ to 6064 ± 1899 mm^3^ (*p* = 0.010). Total airway volume decreased from 25,741 ± 9195 mm^3^ to 22,164 ± 6262 mm^3^; however, this change did not reach statistical significance (*p* = 0.075). PAS values and minAx significantly decreased postoperatively in the double-jaw group (*p* = 0.002 and *p* = 0.001, respectively). Similarly, in the single-jaw group, oropharynx volume decreased significantly from 15,620 ± 5040 mm^3^ to 12,444 ± 4701 mm^3^ (*p* = 0.010), while nasopharynx volume did not show a significant change (*p* = 0.551). Total airway volume significantly decreased from 20,452 ± 7754 mm^3^ to 16,846 ± 6529 mm^3^ (*p* = 0.010), and PAS and minAx exhibited significant postoperative reductions (*p* = 0.000 for both). In addition to statistical significance, Cohen’s *d* values were calculated to assess the clinical relevance of volumetric changes. In the single-jaw group, a large effect was observed for PAS (*d* = 1.213), while moderate effects were noted for oropharynx volume (*d* = 0.652), total airway volume (*d* = 0.503), and minAx (*d* = 0.633). In the double-jaw group, the reduction in PAS also demonstrated a large effect (*d* = 0.866), whereas changes in oropharynx and total airway volumes showed moderate effects (*d* = 0.488 and 0.455, respectively). An increased nasopharyngeal volume in the double-jaw group showed a small-to-moderate effect size (*d* = 0.389).

Table 3 compares the postoperative changes in pharyngeal airway dimensions between the double-jaw and single-jaw surgery groups revealed no statistically significant differences across all measured variables when applying a significance threshold of *p* < 0.01. Although the nasopharyngeal volume exhibited a relatively greater increase in the double-jaw group compared to a decrease in the single-jaw group, this difference did not reach statistical significance (*p* = 0.036). Similarly, reductions in oropharyngeal and total airway volumes, posterior airway space (PAS), and minimum axial area (minAx) were observed in both groups, yet these changes did not differ significantly between them (*p* > 0.01 in all comparisons).

Table 4 shows correlations between the difference in the total airway volume, PAS, and minAx. The analysis demonstrated a significant positive correlation between minAx and PAS (r = 0.427, *p* < 0.01), as well as minAx and total airway volume (r = 0.457, *p* < 0.01). These results indicate that decreases in the minimum airway area are associated with a decrease in both PAS and total airway volume. However, no significant correlation was found between PAS and the total airway volume (r = 0.267, *p* = 0.218). This suggests that while both minAx and PAS are related, and minAx is associated with the total airway volume, PAS alone may not be a reliable predictor of total airway volume in this sample.

## 4. Discussion

Orthognathic surgical procedures affect not only the craniofacial hard tissues and soft-tissue profile but also the size and position of all soft tissues associated with the maxilla and mandible. Based on the direction and extent of skeletal movements, this results in changes in the size of the nasal and oral cavities, as well as the PAS dimensions [12]. It has been reported that mandibular advancement significantly increases the pharyngeal airway, whereas mandibular setback reduces the airway area [12,34,35]. The narrowing of the airway predisposes to respiratory problems and sleep disorders such as OSA [22]. Anatomical risk factors in orthognathic surgery for OSA are the downward movement of the hyoid bone, posterior movement of the base of the tongue, and consequent narrowing of the pharyngeal airways [12]. Maxillomandibular advancement surgeries effectively eliminate OSA by stretching the velopharyngeal and suprahyoid muscles, as well as widening the airway space [36]. In contrast, it is suggested that the movement of the mandible in the posterior direction narrows the airways [22,37].

In the literature, it has been shown that the development of the upper airway is correlated with the developmental period of somatic growth, exhibiting morphological changes as well as volume and dimensional increases [38]. Previous studies have also reported that the upper airway volume is relatively stable at 20 years of age and decreases at an increasing rate after 40 years of age [39]. The patients included in our study consisted of individuals with a mean age of 21.3 years and physiologically stable airway development in accordance with these data.

The posterior airway space is divided into three regions: the nasopharynx, oropharynx, and hypopharynx. However, there is no consistency regarding the terminology and the boundaries of these regions [20]. The various limitations seen in studies make it difficult to evaluate and compare the measurements. In this study, the boundaries of OP and NP volumes were determined based on the study by El and Palomo [40]. The authors stated that measuring the volume of the nasopharynx is more difficult than measuring OP volumes due to the anatomy of the nasal concha region. Therefore, we modified the upper limit of the NP to allow for accurate pre- and post-treatment comparisons using a fixed reference site on the skull such as the Sella point. The program enables users to sculpt the desired airway volume from the 3D structure by adjusting brightness, opacity, and threshold values to remove unwanted voxels and obtain a solid airway volume. In this study, all pharyngeal segments were evaluated separately using this process, allowing for a detailed and region-specific volumetric assessment.

Studies comparing the double-jaw and single-jaw surgery reported that nasopharyngeal dimensions remain unaffected in patients undergoing only the mandibular setback procedure [25,41,42,43]. It is an expected result that the nasopharynx is not affected after mandibular setback surgery alone because anatomically the posterior movement of the mandible occurs anatomically at a distance from the nasopharyngeal region. In contrast, maxillary advancement surgery tends to increase the nasolabial angle, resulting in more favorable air volumes entering the nasal cavity and lower respiratory resistance. In the present study, the nasopharynx volume remained unchanged in the single-jaw group, whereas a statistically significant increase was noted in the double-jaw group.

It is known that isolated mandibular setback surgery decreases the airway volume, whereas bimaxillary surgery causes a limited decrease in the upper airway. This was observed in comparison to mandibular setback alone [44]. In a study by Chen et al. [12] on airway changes after orthognathic surgery, 66 female patients who underwent mandibular setback and maxillary advancement or mandibular setback only were analyzed. They used cephalometric radiographs for evaluating the changes in the airway dimensions. Significant postoperative narrowing of the oropharynx and hypopharynx was only observed in the mandibular setback group. In the group that underwent both maxillary advancement and mandibular setback, there was a transient increase in the nasopharynx and oropharynx regions and a decrease in the hypopharynx region. In the long term, maxillary advancement and mandibular setback surgery resulted in limited change in the pharyngeal airway and this is thought to be due to the advancement of the velopharyngeal muscle system induced by Lefort I osteotomy. Park et al. [7] evaluated the volumetric changes in patients who underwent bimaxillary surgery or isolated mandibular setback surgery. They found that oropharyngeal and hypopharyngeal airways decreased after 4.6 months in the mandibular setback group, and the oropharyngeal airway also decreased. Khaghaninejad et al. [35] conducted a study to compare pharyngeal airway changes following different orthognathic procedures in Class III patients using CBCT analysis. A total of 48 patients were divided into three equal groups undergoing mandibular setback, bimaxillary surgery, or maxillary advancement. CBCT scans were taken immediately preoperatively, one day postoperatively, and six months later. The study found that while all groups experienced an initial reduction in airway volume immediately after surgery, the maxillary advancement group showed a significant increase in airway dimensions at six months compared to baseline. In contrast, the mandibular setback and bimaxillary surgery groups demonstrated persistent reductions. The greatest narrowing occurred after mandibular setback alone. In the present study, although the differences in oropharynx volumes before and after surgery were not significant between the groups, a more pronounced reduction was observed in the single- jaw group. As expected, the decrease in the oropharynx in the single-jaw group was higher than that in the double-jaw group in the present study. By contrast, Jakobsone et al. [28] stated that maxillary advancement of more than 2 mm could increase the volume of the nasopharynx. Unlike many studies in the literature [7,45,46,47], a maxillary advancement of more than 5 mm was performed in our study. Therefore, a remarkable increase in the nasopharynx occurred. This increase in the nasopharynx might offset the alteration of the oropharynx from a mandibular setback in the double-jaw surgery.

There is no evident correlation between pharyngeal airway volume and respiratory function [31]; therefore, airway reduction does not suggest a greater risk for OSA. Uesugi et al. [43] reported no difference in the apnea–hypopnea index between single- and double-jaw surgery. However, they pointed out that it is important to assess each case individually. Studies involving patients with sleep apnea revealed that maxillomandibular advancement surgeries enhanced sleep quality and decreased the apnea–hypopnea index [36,48]. Some recent studies have adopted a multidisciplinary approach by combining volumetric airway analysis with polysomnography (PSG) and subjective sleep assessments. For example, On et al. [13] reported decreases in PAS and airway volume alongside worsened PSG indices after mandibular setback in Class III patients. Kongsong et al. [49] reported that decreased airway volume and the minimum constricted areas following mandibular setback surgery. Additionally, they noted that sleep quality initially worsened but showed improvement after one year. On the other hand, Scherer et al. [50] reported that patients with Class III malocclusion who underwent bimaxillary orthognathic surgery did not have an increased risk of developing obstructive sleep apnea or a decline in sleep-related quality of life compared to those treated with orthodontics alone. An opening in the nasopharyngeal airway in the double-jaw group might be expected to improve nasal breathing.

When the PAS and minAx measurements were examined, a decrease was observed in both groups. In addition to variations in total airway volume reduction, recent studies have reported that a negative change in the minimum axial cross-sectional area is a significant risk factor for OSA, highlighting a correlation between the minAx and total airway volume [51,52,53]. Our results agree with the literature; while both minAx and PAS are related, and minAx is associated with the total airway volume, PAS alone may not be a reliable predictor of total airway volume in this manner. In a CT study involving patients with various degrees of OSA, including a control group, Avraham et al. showed that the minimum axial area is 50 mm^2^ or less in patients with severe OSA, 60–100 mm^2^ in patients with moderate OSA, and 110 mm^2^ in the control group [54]. In our study, although there was a greater decrease in the minAx was observed in the double-jaw group, the mean value did not fall below 100 mm^2^ in both groups. By contrast, Abramson et al. [55] stated that the only correlation between CT parameters and cephalometric measurements is the PAS value. As such, the question arises as to whether a constricted section at the base of the airways could pose an issue despite adequate overall airway volume. Partinen et al. [56] reported that a posterior airway space of less than 5 mm causes severe respiratory problems. In our study, although the mean postoperative PAS values were above 6 mm in both groups, a significant decrease in airway volume was observed. This finding is consistent with previous studies that have recorded the notable narrowing of the PAS after mandibular setback [37,57,58].

After orthognathic surgery, the airways are immediately affected and undergo further changes over time. Several studies have shown that airway dimensions, especially in the oropharyngeal region gradually increase over time compared to short-term results [7,45,46]. Kang et al. [59] reported that, in patients with Class III malocclusion, mandibular setback surgery alone caused a greater reduction in the pharyngeal airway than bimaxillary surgery during the first postoperative year, and they stated that bimaxillary surgery is more stable in regard to the airway. However, it has been reported that even 6 years after surgery, the original airways are not fully restored [47]. We evaluated that a follow-up period of at least 3 months may not be sufficient to evaluate long-term changes, despite adequately showing for the improvement of hypertrophy occurring in soft tissues after surgery [60].

One of our study limitations is the small sample size. Nevertheless, we carefully selected patients based on our inclusion criteria to ensure the homogeneity of the study group. Patients requiring severe maxillary impaction or asymmetrical mandibular movement were not included. Another important limitation is the absence of functional respiratory assessments before and after surgery. Although volumetric changes suggest possible effects, it is unclear whether these lead to actual improvements or problems in breathing problems. Additional studies with larger sample sizes and long-term follow-up at multiple postoperative time points are necessary to determine whether the observed increases in airway volume are stable or transient. These studies should also incorporate individual-level assessments, comparisons of different surgical protocols, evaluations of soft tissue behavior, and the integration of volumetric and functional respiratory analyses to comprehensively assess airway changes following orthognathic surgery.

## 5. Conclusions

This study demonstrated that orthognathic surgery exerts a differential impact on the pharyngeal airway depending on the surgical approach. In the double-jaw group, a significant increase in nasopharyngeal volume was observed following surgery, whereas the reduction in oropharyngeal volume did not reach statistical significance. Conversely, in the single-jaw group, oropharyngeal volume significantly decreased postoperatively, while nasopharyngeal volume remained unchanged. Moreover, total airway volume showed a statistically significant reduction only in the single-jaw group. Importantly, both groups exhibited a marked decrease in the PAS and minAx value, suggesting a considerable influence on posterior airway morphology. From a clinical perspective, bimaxillary surgery may offer advantages in preserving or increasing airway volume, particularly in the nasopharyngeal region. Therefore, in patients with narrow pharyngeal airways or a predisposition to sleep-disordered breathing, a two-jaw surgical approach might be preferable.

## Figures and Tables

**Figure 1 diagnostics-15-02217-f001:**
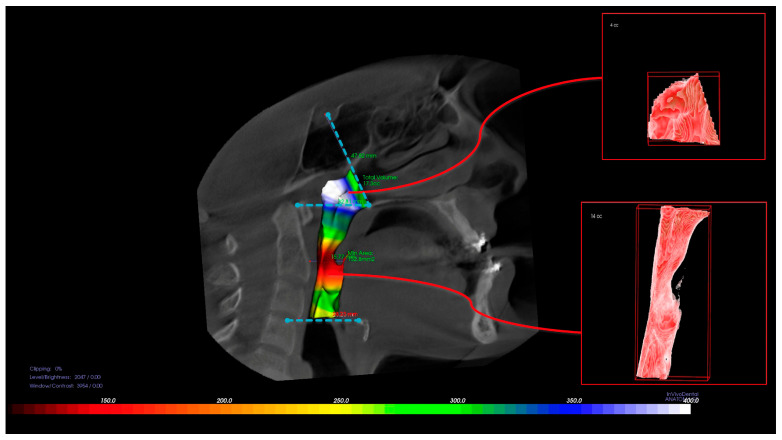
Three-dimensional representation of the nasopharyngeal (NP) and oropharyngeal (OP) airway segmentation based on CBCT images. The NP region extends from the posterior nasal spine to the tip of the soft palate, while the OP region spans from the soft palate to the level of the epiglottis.

**Figure 2 diagnostics-15-02217-f002:**
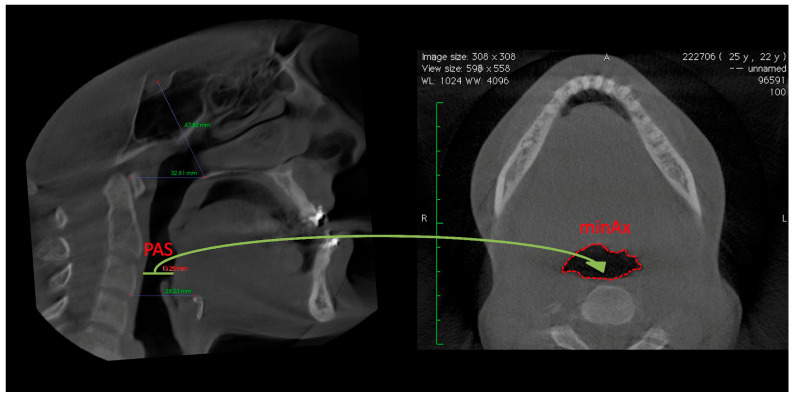
Measurement of the posterior airway space (PAS) and minimum axial area (minAx) on sagittal CBCT slices. The posterior airway space (PAS) was measured as the narrowest anteroposterior distance between the base of the tongue and the posterior pharyngeal wall. The minimum axial area (minAx) was determined as the smallest cross-sectional area of the airway, orthogonal to its curved axis, and represents the most constricted region.

**Figure 3 diagnostics-15-02217-f003:**
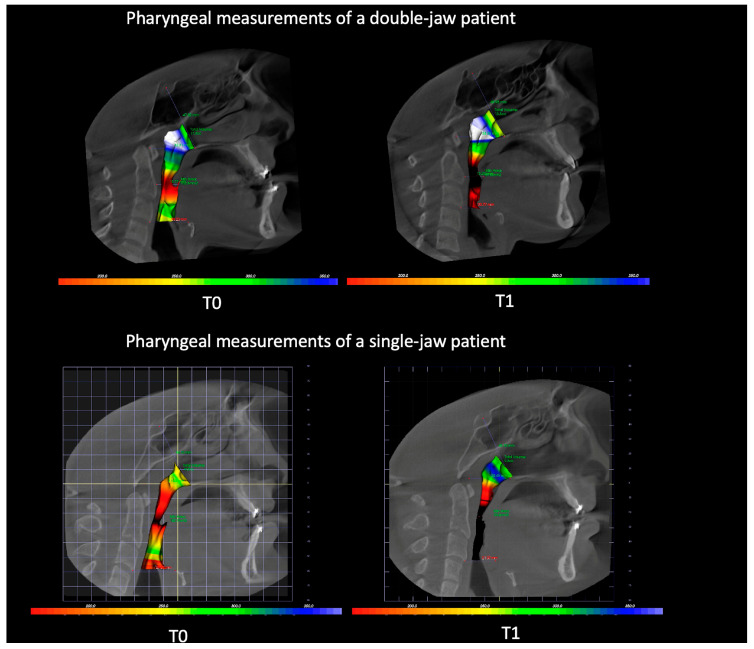
Comparison of preoperative and postoperative airway volumes using 3D reconstructed CBCT data in a representative patient from each group. The upper image represents a patient from the double-jaw group, while the lower image shows a patient from the single-jaw group. Changes in airway dimensions are clearly noticeable.

**Table 1 diagnostics-15-02217-t001:** Amount of movement (mm) of maxilla and mandible in double- and single-jaw group (*p* ≤ 0.01).

Amount of Movement (mm)	Maxilla	Mandible	*p*
Double jaw	5.28 ± 2.07	7.84 ± 2.43	0.339
Single jaw	-------	7.52 ± 1.2	

**Table 2 diagnostics-15-02217-t002:** Volume of oropharynx, nasopharynx, and total airway before and after surgery.

	Double Jaw				Single Jaw			
	T1	T2	*p*	*d*	T1	T2	*p*	*d*
Oropharynx volume (mm^3^)	17,097 ± 5675	14,290 ± 5835	0.017	0.488	15,620 ± 5040	12,444 ± 4701	0.010 *	0.652
Nasopharynx volume (mm^3^)	5316 ± 1948	6064 ± 1899	0.010 *	0.389	4563 ± 2125	4693 ± 2201	0.551	0.06
Total airway volume (mm^3^)	25,741 ± 9195	22,164 ± 6262	0.075	0.455	20,452 ± 7754	16,846 ± 6529	0.010 *	0.503
PAS (mm)	9.60 ± 4.17	6.43 ± 3.07	0.002 *	0.866	9.68 ± 2.69	6.50 ± 2.55	0.000 *	1.213
minAx (mm^2^)	189 (77.9–300)	100 (36.9–236.8)	0.001 *	--	168.02 ± 78.51	119.13 ± 75.93	0.000 *	0.633

* *p* ≤ 0.01; Median (min–max) is given for the nonparametric test. Cohen’s *d* values are reported to indicate the effect size for volumetric differences. Effect sizes were interpreted as small (*d* = 0.2), medium (*d* = 0.5), or large (*d* = 0.8).

**Table 3 diagnostics-15-02217-t003:** Comparisons of the difference between T2 and T1 according to the surgery procedure.

	Double Jaw	Single Jaw	*p*
∆OP (OP2 − OP1)	−2762 ± 3772	−3408 ± 3343	0.674
∆NP (NP2 − NP1)	707 ± 922	−197 ± 1009	0.036
∆TOTAL (TOTAL2 − TOTAL1)	−2055 ± 4233	−3605 ± 3823	0.375
∆PAS (PAS2 − PAS1)	−3.18 ± 2.9	−3.17 ± 1.6	0.997
∆minAx (minAx2 − minAx1)	−71.8 (−172.8–175.72)	−44.3 (−81.9–17.1)	0.239

Median (min–max) is given for the nonparametric test.

**Table 4 diagnostics-15-02217-t004:** Spearman correlations between the difference in total airway volume, PAS, and minAx.

		∆minAx	∆PAS	∆TOTAL
**∆minAx**	Correlation Coefficient	1	0.427 *	0.457 *
	*p*		0.042	0.029
	*N*	23	23	23
**∆PAS**	Correlation Coefficient	0.427 *	1	0.267
	*p*	0.042		0.218
	*N*	23	23	23
**∆TOTAL**	Correlation Coefficient	0.457 *	0.267	1
	*p*	0.029	0.218	
	*N*	23	23	23

* Correlation is significant at the 0.05 level (2-tailed).

## Data Availability

The data presented in this study are available on request from the corresponding author due to privacy.

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
