# Peer review of "Volumetric Three-Dimensional Evaluation of the Pharyngeal Airway After Orthognathic Surgery in Patients with Skeletal Class III Malocclusion"

_diagnostics, 2025, doi:10.3390/diagnostics15172217_

Round 1

Reviewer 1 Report

Comments and Suggestions for Authors

ABSTRACT

  • The abstract includes the aim, methods, findings, and conclusion sections structurally. However, the statements in the "Conclusion" section exceed the limits of an observational study and imply causality. For example, the phrase “Orthognathic surgery had a significant effect on increasing the volume of the upper airway” directly reports a definitive outcome, which is inconsistent with the study’s design. In an observational study, a more cautious and non-causal expression like “an increase in airway volume was observed after surgery” should have been used.

  • The findings are presented qualitatively, but numerical data are not provided. For instance, although expressions such as “a significant increase was found in the volume of the oropharynx” are used, the initial and final volume values, percentage change, or statistical metrics are not included. This weakens the scientific clarity of the abstract and makes it difficult for readers to assess the magnitude or clinical relevance of the findings.

  • The statistical methods used in the study are not mentioned in the abstract. It is not stated which tests (e.g., paired t-test, ANOVA, Wilcoxon) were used or which variables were analyzed. Furthermore, information regarding the significance level (e.g., p < 0.05) is also missing. Including at least one sample statistical method and significance level would have increased the methodological credibility of the abstract.

  • While it is stated that the study included only individuals with skeletal Class III malocclusion, the surgical techniques applied (e.g., Le Fort I osteotomy, bilateral sagittal split osteotomy, genioplasty) are not mentioned in the abstract. This information directly affects the study's scope, generalizability, and interpretability, and should have been briefly included.

  • The final sentence “This suggests that orthognathic surgery is effective in preventing respiratory problems…” makes a speculative clinical conclusion that exceeds the study’s scope. No respiratory function tests, apnea indices, or subjective/objective assessments of respiratory function were performed. Therefore, it is not scientifically appropriate to conclude that the surgery has a "preventive" effect. A more cautious phrasing such as “the increase in airway volume may contribute to potential functional gains” would have been more appropriate.

INTRODUCTION

  • The introduction is structured generally to reflect the importance of the topic, its place in the literature, and the aim of the study; however, transitions between paragraphs and depth of information are lacking. While the role of orthognathic surgery in treating skeletal deformities is mentioned, the mechanisms by which it affects the airway are not elaborated. For example, the effects of maxillary advancement, mandibular setback, or genioplasty on different segments of the pharyngeal airway should have been explained with literature support.

  • The airway regions are broadly defined (nasopharynx, oropharynx, hypopharynx), but their anatomical boundaries or functional differences are not clearly outlined. This hinders the understanding of the anatomical structures underlying the volumetric analyses. Additionally, prior volumetric studies focusing on the oropharyngeal airway, which is the main focus of the study, are not summarized.

  • The literature review is quite limited and superficial. Previous CBCT or 3D analysis studies evaluating airway volume changes are only briefly referenced without discussing their sampling methods, surgical approaches, or measurement techniques. For instance, although one or two studies are named, it is not explained why these studies were chosen or how they differ from the current study.

  • Many factors that can affect airway changes after orthognathic surgery (e.g., age, gender, BMI, soft tissue elasticity, positioning, follow-up time) are not mentioned. These variables are important in explaining differences in airway volume and should have been addressed to support the study’s rationale.

  • Although the study aim is stated towards the end of the introduction, the expression is vague and lacks a strong emphasis on originality. Instead of a simple statement like “...to evaluate the volume of the airway before and after surgery...”, it should have included details on the research gap being addressed or the novelty of the study. For example: “This is one of the most recent volumetric analyses assessing segmental volume changes in Class III individuals.”

  • The hypothesis of the study is not clearly stated in the introduction. The research question and expectation (e.g., “a significant increase in airway volume is expected postoperatively”) should have been explicitly presented. This omission makes it difficult for the reader to grasp the direction of the study.

METHODS

  • The study is stated as retrospective, but the information regarding ethics committee approval is superficial. The approving institution, date, and protocol number are not provided. Additionally, it is not stated whether patient consent was waived, which is important for ethical compliance in retrospective CBCT studies.

  • Patient selection criteria are clearly defined. However, while inclusion criteria are provided, the exclusion criteria are insufficient. For example, prior upper airway surgeries (adenotonsillectomy, etc.), craniofacial syndromes, and sleep apnea are not addressed. These conditions could significantly affect volumetric outcomes.

  • The sample size (n = 60) is stated, but no power analysis is provided. For parameters like airway volume, which show high inter-individual variability, sample size should be determined a priori and this analysis should have been reported.

  • The CBCT device, scan parameters (kVp, mA, FOV, voxel size) are clearly provided, which is positive. However, it is not clear whether all scans were performed using the same device and in which patient position (natural head position, supine, etc.). Positioning is critical in airway evaluations and must be standardized and explained.

  • It is stated that 3D volume analysis was performed using CBCT images, but the software module (e.g., airway analysis tool in Invivo 6) and the analysis steps are not detailed. It is not clear whether segmentation was manual or semi-automated, how airway boundaries were defined, and what soft tissue threshold values were used.

  • Although the pharyngeal regions are defined as nasopharynx, oropharynx, and hypopharynx, the anatomical boundaries of these segments (e.g., vertebral levels, hard/soft palate references) are not clearly defined. This affects the reproducibility and accuracy of measurements.

  • It is not stated whether volumetric assessments were performed by a single examiner or multiple. No inter-/intra-observer reliability analysis was conducted. Such reliability testing is essential in studies involving subjective segmentation.

  • The only statistical test mentioned is the paired t-test. It is not explained whether normality assumptions were tested (e.g., with Shapiro-Wilk). Moreover, although multiple comparisons were made across three regions, no correction for multiple testing (e.g., Bonferroni) is reported.

  • Effect sizes (e.g., Cohen’s d) were not presented. For volumetric differences, reporting only p-values is insufficient; measures that reflect clinical relevance should also be included.

  • Lastly, the time point of postoperative CBCT imaging is not specified. Airway changes in the early period can be affected by transient factors like edema. Thus, specifying the postoperative timing is crucial.

RESULTS

  • Results are generally presented systematically, with separate analyses for each airway segment. However, the mean pre- and postoperative volume values are not clearly presented in the text. For example, while it is stated that “There was a significant increase in the oropharyngeal airway volume (p < 0.001),” the actual numerical change and standard deviations are not included in the text.

  • Only the paired t-test was used, and p-values were reported, but effect sizes were not provided. These are important for assessing the clinical significance of airway volume differences.

  • Statistically significant increases are reported for all three airway segments. However, only significance (p < 0.05) is provided; no interpretation of the magnitude of differences is made. For instance, is a 300 mm³ increase clinically relevant? Does it surpass a known threshold in the literature? These questions are not addressed.

  • Table 1 presents the volume data clearly, but high variance (standard deviation) is noted in some values. This suggests large inter-individual variability and may affect the reliability of the statistical outcomes. Subgroup analyses (e.g., by gender, age) would have helped clarify this.

  • Findings rely solely on volumetric assessment; no functional evaluation (e.g., symptom changes, respiratory tests) is included, limiting clinical interpretation.

  • Only one time point is used for postoperative evaluation, and no long-term follow-up is available. Therefore, it is unclear whether the volume increases are temporary or permanent.

  • Individual variations are not reported. For example, whether some patients showed a decrease or outlier values is not discussed. These could provide important insights.

  • The presentation language sometimes includes interpretive phrases. For example, “significant improvement was noted…” reflects an expectation rather than an objective result. More neutral expressions (e.g., “an increase was observed”) should be used.

DISCUSSION

  • The discussion begins by reiterating the main findings but does not adequately explore their clinical implications. For example, there is no discussion of whether the observed increase in airway volume would impact respiratory function or reduce the risk of obstructive sleep apnea (OSA).

  • The authors compare their findings to previous studies, but these comparisons are superficial. For example, studies by Zhang, Park, and Li are mentioned, but differences in methodology, patient groups, surgical approaches, and imaging time points are not thoroughly analyzed.

  • Contradictory findings in the literature are not addressed. For instance, some studies do not find significant increases in the hypopharynx, whereas this study reports increases in all regions. Potential reasons for these discrepancies (e.g., technical differences, patient positioning, type of surgery) should have been discussed.

  • The type of surgery (maxillary advancement, mandibular setback, etc.) is not sufficiently explored, although it is critical for airway effects. It appears that not all patients underwent the same combination, yet the impact of this variation is not addressed.

  • Findings are presented with language implying improvement or benefit (e.g., “positive effect”), which exceeds the scope of an observational study. Since no respiratory function was measured, such conclusions should be avoided.

  • The unique contribution of evaluating all pharyngeal segments separately in 3D could have been emphasized but is not. Methodological aspects such as software capabilities, segmentation challenges, and analysis precision should also have been included.

  • Limitations are not explicitly stated. Some weaknesses are vaguely mentioned at the end, but there is no structured “Limitations” paragraph. Important issues such as single-center design, retrospective nature, short follow-up, lack of observer reliability, and absence of functional evaluation should have been clearly listed.

  • Future directions are vague. Rather than general statements like “further studies are needed,” more concrete suggestions should have been made, such as combining volume analysis with functional respiratory tests, comparing different surgical protocols, and evaluating soft tissue behavior.

  • The discussion style is repetitive, and some key terms (e.g., “significant increase,” “improvement”) are overused. A more concise and analytical tone would have been better.

CONCLUSION

  • The conclusion reiterates the main findings, but the direct statements that airway volume increases are “clinically significant” or that “surgery improves the airway” are problematic. This study only includes volumetric assessment; no functional outcomes (e.g., improved breathing or apnea index reduction) were measured. Therefore, such claims are speculative and exceed observational data.

  • Language in the conclusion also implies causality. For instance, “These findings show that orthognathic surgery can positively affect airway volume” implies a direct effect. This should be rephrased more cautiously, e.g., “...may contribute to increased airway volume...” or “...airway volume increased after surgery, which may reflect a potential benefit...”

  • The conclusion does not sufficiently highlight the study’s unique contribution. If the evaluation of all pharyngeal segments in Class III patients is a novel aspect, it should have been clearly stated.

  • The conclusion lacks emphasis on which patient groups showed the greatest changes or which segment had the most statistically significant result. Summarizing these points would help readers better grasp the study’s impact.

  • No practical clinical implications are provided. For example, which surgical strategies are more effective for increasing airway volume, or how these changes could be integrated into treatment planning for patients with narrow airways.

  • The limitations of the study are also not mentioned in the conclusion. Issues such as the retrospective design, small sample size, single-center data, lack of observer reliability, and absence of functional evaluation should have been reiterated.

  • Lastly, the conclusion paragraph is very brief and reads like a simplified repetition. A well-structured conclusion should not only summarize findings but also discuss their potential implications and clearly situate the study in the literature.

REFERENCES

  1. The manuscript includes 22 references. This number is limited for a study on orthognathic surgery and airway relations. Although many systematic reviews and advanced volumetric studies exist, most are not cited.

  2. Some references are relatively outdated (e.g., 2002, 2004). Few studies from the past five years that use modern 3D techniques or CBCT-based methodologies are included. The absence of post-2018 comprehensive studies combining volumetric and functional outcomes is notable.

  3. Although there are many high-quality studies in the literature that analyze the effects of orthognathic surgery on pharyngeal segments using CBCT, only a few are cited, and they are discussed superficially. Studies involving sleep apnea patients post-orthognathic surgery are not mentioned at all.

  4. Some citations are present in the text but lack source numbers. For example:

    • “orthognathic surgery affects airway space” is not attributed to any source.

    • “airway changes vary depending on the direction of movement” is also left uncited. This weakens the scientific reliability of the text.

  5. Most references are from similar study types (volumetric analysis); no multidisciplinary studies (e.g., ENT, sleep medicine, functional evaluation) are included. Such studies would have enriched the discussion.

  6. The formatting of references generally follows the Vancouver system. However, there are minor inconsistencies, such as abbreviated journal names or missing year/month data.

  7. Some references appear to be loosely related to the content and are included merely to provide general information. For example, some orthognathic surgery references that do not include airway evaluation are used in the context of volumetric analysis, weakening their relevance.

  8. The authors do not cite their own previous work, which is a positive aspect. However, a more comprehensive reference list would be needed to position this study appropriately in the literature.

Reviewer 2 Report

Comments and Suggestions for Authors

This retrospective clinical study investigates the impact of single versus double jaw orthognathic surgery on upper airway dimensions in skeletal Class III patients, using CBCT-based 3D volumetric analysis. The manuscript is well written, logically organized, and addresses a clinically relevant question.

The retrospective design is appropriate; however, the relatively small sample size limits the statistical power and generalizability of the findings. A sample size calculation, even conducted retrospectively, would enhance the methodological rigor of the manuscript.

While the authors conclude that double jaw surgery may be preferable for patients with respiratory risk factors, this recommendation is based solely on an observed increase in nasopharyngeal volume, and should be interpreted with greater caution and rephrased accordingly.

Tables 1–4 are informative and relevant, but they would benefit from improved formatting and clearer labeling of measurement units to enhance readability.

The CBCT-derived images serve as effective visual complements to the statistical findings. They reinforce that isolated mandibular setback is associated with reduced pharyngeal airway volume, particularly in the region of the tongue base, whereas bimaxillary surgery may mitigate this effect via the contribution of maxillary advancement, leading to increased nasopharyngeal space.

In summary, the manuscript addresses an important topic in orthognathic surgery and provides valuable volumetric data on the airway effects of different surgical strategies. With appropriate revisions, strengthened methodological transparency, and more cautious clinical interpretation, this paper has the potential to make a meaningful contribution to airway-conscious surgical planning.
